# A therapeutic neutralizing antibody targeting receptor binding domain of SARS-CoV-2 spike protein

Cheolmin Kim[1,11], Dong-Kyun Ryu [1,11], Jihun Lee [1,11], Young-Il Kim[2,11], Ji-Min Seo[1], Yeon-Gil Kim[3], Jae-Hee Jeong[3], Minsoo Kim[1], Jong-In Kim[1], Pankyeom Kim[1], Jin Soo Bae [1], Eun Yeong Shim[1], Min Seob Lee [1], Man Su Kim[1], Hanmi Noh[1], Geun-Soo Park[1], Jae Sang Park[1], Dain Son[1], Yongjin An[1], Jeong No Lee[1], Ki-Sung Kwon[1], Joo-Yeon Lee[4], Hansaem Lee[4], Jeong-Sun Yang[4], Kyung-Chang Kim[4], Sung Soon Kim[4], Hye-Min Woo[4], Jun-Won Kim[4], Man-Seong Park[5], Kwang-Min Yu[2], Se-Mi Kim[2], Eun-Ha Kim[2], Su-Jin Park[2,10], Seong Tae Jeong[6], Chi Ho Yu[6], Youngjo Song[6], Se Hun Gu[6], Hanseul Oh[7], Bon-Sang Koo[7], Jung Joo Hong[7], Choong-Min Ryu[8], Wan Beom Park[9], Myoung-don Oh[9], Young Ki Choi [2✉] & Soo-Young Lee [1✉]

Vaccines and therapeutics are urgently needed for the pandemic caused by severe acute respiratory syndrome coronavirus 2 (SARS-CoV-2). Here, we screen human monoclonal antibodies (mAb) targeting the receptor binding domain (RBD) of the viral spike protein via antibody library constructed from peripheral blood mononuclear cells of a convalescent patient. The CT-P59 mAb potently neutralizes SARS-CoV-2 isolates including the D614G variant without antibody-dependent enhancement effect. Complex crystal structure of CT-P59 Fab/RBD shows that CT-P59 blocks interaction regions of RBD for angiotensin converting enzyme 2 (ACE2) receptor with an orientation that is notably different from previously reported RBD-targeting mAbs. Furthermore, therapeutic effects of CT-P59 are evaluated in three animal models (ferret, hamster, and rhesus monkey), demonstrating a substantial reduction in viral titer along with alleviation of clinical symptoms. Therefore, CT-P59 may be a promising therapeutic candidate for COVID-19.

[1] Biotechnology Research Institute, Celltrion Inc, Incheon 22014, Republic of Korea. [2] College of Medicine and Medical Research Institute, Chungbuk National University, Cheongju, Republic of Korea. [3] Pohang Accelerator Laboratory, Pohang University of Science and Technology, Pohang, Kyungbuk 790-784, Korea. [4] Center for Infectious Diseases Research, Korea National Institute of Health, Korea Centers for Disease Control and Prevention, Cheongju, Republic of Korea. [5] Department of Microbiology and Institute for Viral Diseases, College of Medicine, Korea University, Seoul, Republic of Korea. [6] The 4th R&D Institute, Agency for Defense Development, Yuseong, P.O.Box 35, Daejeon 34186, Republic of Korea. [7] National Primate Research Centre, Korea Research Institute of Bioscience and Biotechnology, Cheongju, Republic of Korea. [8] Infectious Disease Research Centre, Korea Research Institute of Bioscience and Biotechnology, Daejeon, Republic of Korea. [9] Department of Internal Medicine, Seoul National University College of Medicine, Seoul, Republic of Korea. [10] Present address: Division of Applied Life Science and Research Institute of Life Sciences, Gyeongsang National University, Jinju 52828, Korea. [11] These authors contributed equally: Cheolmin Kim, Dong-Kyun Ryu, Jihun Lee, Young-Il Kim. ✉email: choiki55@chungbuk.ac.kr; sooyoung.lee@celltrion.com

Severe acute respiratory syndrome coronavirus 2 (SARS-CoV-2) is a novel virus that has rapidly spread to pandemic status, resulting in an unprecedented global health and economic crisis. Furthermore, vaccines and effective treatments are not currently available. Convalescent plasma treatment including neutralizing antibody could inhibit viral replication and alleviate severe clinical symptoms[1-3]. Due to their mode of action, neutralizing antibodies may be ideally suited for administration in combination with other classes of antiviral therapy, such as remdesivir. However, convalescent plasma treatment is practically limited due to lack of scalability and lot-to-lot variation. Therefore, the monoclonal antibody would be a good alternative to meet unmet medical demands therapeutically and prophylactically.

As with SARS-CoV-1, causing an epidemic in 2002, SARS-CoV-2 was defined to utilize its own S protein to interact with ACE2 as a functional receptor for viral entry[4-6]. The S1 subunit-containing RBD is responsible for viral attachment and entry, while the S2 subunit mediates cell membrane fusion following proteolytic activation[6]. Initial studies showed that the RBD can binds to ACE2 with higher affinity than that of SARS-CoV-1, partly demonstrating rapid global transmission and pathogenesis[6,7]. Indeed, over time, SARS-CoV-2 isolates bearing D614G mutations in the viral S protein variant have emerged that enable more efficient cellular entry and ultimately lead to an enhanced viral transmission. This variant is now known to be dominant in many countries, especially in Europe and the United States[8-10]. RBD is a key determinant for viral replication and also is known to be an immunological main target for SARS vaccines with little or no antibody-dependent enhancement (ADE)[11].

Here, we report a mAb, CT-P59, as a strong binder for SARS-CoV-2 RBD. CT-P59 inhibits SARS-CoV-2 infection via steric hindrance with ACE2 receptor and mitigates the infection symptoms both in vitro and in vivo. CT-P59 mAb, along with small molecule drugs such as remdesivir and dexamethasone, may thus help curb pandemic as a therapeutic or preventative intervention for COVID-19.

## Results

**Screening and characterization of CT-P59**. To identify novel SARS-CoV-2-targeting neutralizing antibodies, we isolated RBD-binding single-chain variable fragments by utilizing recombinant SARS-CoV-2 RBD as bait for phage display screening. A monoclonal antibody reformatted to fully human immunoglobulin (IgG), termed CT-P59, was assessed for its neutralization potency by in vitro plaque reduction neutralization test (PRNT) against authentic SARS-CoV-2 and SARS-CoV-2 D614G variant (Fig. 1a). CT-P59 was shown to significantly inhibit viral replication with the value of low half-maximal inhibitory concentration ($IC_{50}$) (8.4 ng/ml) against a SARS-CoV-2 clinical isolate in Korea, which showed identical genome sequence of S protein with the primary virus in China (Accession ID: YP_009724390.1). We found that CT-P59 reduced the replication of the D614G variant with the value of $IC_{50}$ (5.7 ng/ml) to a similar extent as the wild-type virus. In addition, a competitive binding assay with biolayer interferometry (BLI) revealed that CT-P59 completely inhibited the binding of RBD-ACE2 (Fig. 1b). In parallel, we carried out the RBD-binding and ACE2 interference test with RBD mutant proteins which were reported and commercially available[9,12,13]. We found that CT-P59 can bind to these mutants and completely inhibit binding between ACE2 and RBD mutants by BLI (Fig. 1b and Supplementary Table 1). Furthermore, CT-P59 binding specificity to other coronaviruses (SARS-CoV, HCoV-HKU1, and MERS-CoV) was evaluated by BLI, indicating that CT-P59 can bind specifically to SARS-CoV-2 (Supplementary Fig. 1). Next,

surface plasmon resonance analysis demonstrated that CT-P59 has a high affinity for SARS-CoV-2 RBD with a $K_D$ value of 27 pM (Supplementary Fig. 2).

**Structural basis of neutralization**. To investigate the neutralizing mechanism of CT-P59, the crystal structure of the CT-P59 Fab/SARS-CoV-2 RBD complex was determined using X-ray crystallography at 2.7 Å resolution (Supplementary Table 2). The complex structure shows that CT-P59 binds to the receptor-binding motif (RBM) within SARS-CoV-2 RBD, which directly interacts with ACE2 (Fig. 2a). The association angle between CT-P59 and the RBD is different from that reported for other structure available neutralizing antibodies in complex with the RBD (Fig. 2b and Supplementary Fig. 3a)[14-25]. These observations indicate that the epitopes of CT-P59 are distinct from those of other antibodies (Fig. 2c). Further, the interactions of the RBD with the heavy and light chains of CT-P59 bury a solvent-accessible surface area of 825 and 113 Å[2], respectively, calculated by PISA[26]. Consistently, most of the interaction between the two proteins is mediated by the heavy chain involving all three complementarity determining regions (CDRs). In total, 16 residues from the CT-P59 heavy chain interact with 19 residues of the RBD at a distance cutoff of 4.5 Å (Supplementary Table 3). Of note, the β-hairpin structure of the CDR H3 of 18 amino acids plays a crucial role in the strong association with the RBD, by forming eight hydrogen bonds as well as hydrophobic interactions involving several of aromatic residues in the middle of the ACE2-binding surface (Fig. 2d). The light chain shows marginal contact with the RBD involving parts of CDR L1 and L2 where only three residues interact with four residues of the RBD (Supplementary Table 3).

To further analyze the structural basis for blocking of the interaction between RBD and ACE2 by CT-P59, the complex structure of CT-P59-RBD was superimposed on the RBD-ACE2 structure (PDB 6LZG)[27]. CT-P59 binding does not alter the overall conformation of the RBD structure in which the pairwise root-mean-square deviation between the Cα atoms of the two RBD structures is 0.89 Å over 193 atoms. However, the β5–β6 loop region (residues 473–488) of the RBD shows a local conformational change, which might be induced by the interaction with CT-P59. The structural superposition reveals that the heavy chain of CT-P59 overlaps completely with ACE2 protein, while the light chain overlaps partially with the receptor (Supplementary Fig. 4a). In agreement with the superposition, there is a substantial overlap between the CT-P59 and ACE2-binding surface areas on RBD (Supplementary Fig. 4b). Among the 21 residues of RBD that interact with ACE2, 12 are also involved in the interaction with CT-P59, when a distance cutoff of 4.5 Å is applied (Fig. 2c). These observations indicate that the binding of CT-P59 to RBD directly occludes the binding surface of ACE2.

**In vivo efficacy in animal models**. To demonstrate in vivo antiviral efficacy of CT-P59 in terms of viral clearance and clinical symptoms, viral loads, and lung pathology, we conducted virus challenge studies employing three animal models (ferrets, golden Syrian hamsters, and rhesus monkeys). In the ferret study, the virus was challenged via both intranasal and intratracheal routes, followed by intravenous treatment of CT-P59 and isotype control at 1 day post-infection (dpi). The animals given 30 mg/kg and 3 mg/kg showed significantly reduced viral RNA from 2 dpi and 4 dpi, respectively. At 2 dpi, the infectious virus titer ($TCID_{50}$) in the nasal wash was significantly decreased in animals treated with 30 mg/kg of CT-P59 when compared to controls, and the infectious virus was not detected at 6 dpi (Fig. 3a, b). The animal with

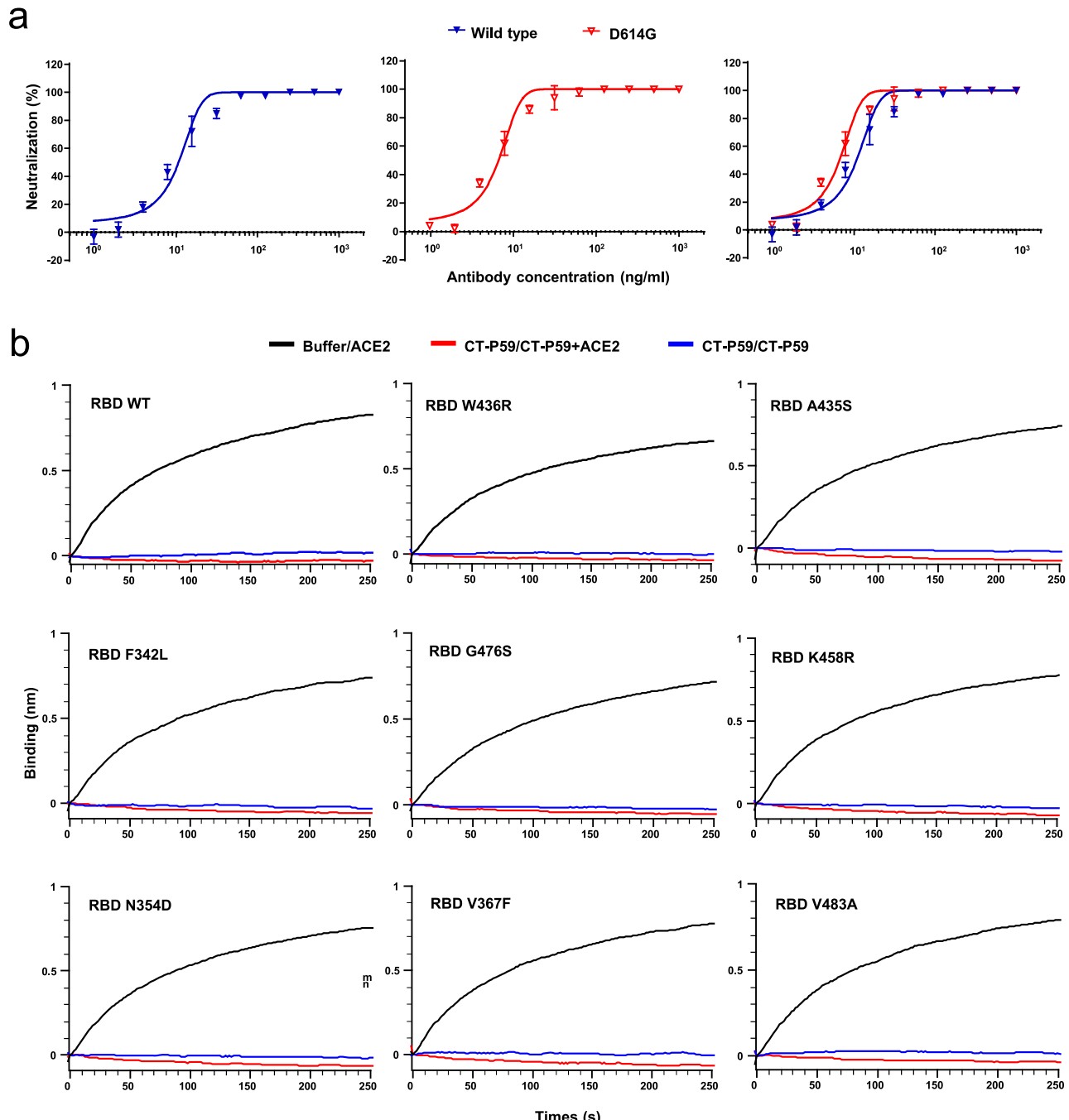

**Fig. 1 CT-P59 can effectively neutralize SARS-CoV-2 in vitro by blocking RBD-ACE2 binding. a** Serial twofold-diluted CT-P59 were incubated with SARS-CoV-2 live viruses; wild type (blue) and D614G (red). The mixture was added to VeroE6 cells. After 2–3 days of incubation, the neutralization activity was evaluated by counting plaques. Two independent experiments were performed in duplicate. **b** SARS-CoV-2 RBD immobilized on biosensor was saturated with CT-P59. Then, CT-P59 flowed over the biosensor surface in the presence (red) or absence (blue) of the ACE2 receptor. As a positive control, the buffer was loaded onto SARS-CoV-2 RBD immobilized biosensor, and ACE2 flowed over the biosensor surface (black).

30 mg/kg also showed the decreased viral RNA at 3 dpi in the lungs. Further, the infectious virus titer was significantly attenuated in lung tissues with both doses at 3 dpi and not detectable with 30 mg/kg at 7 dpi (Fig. 4a, d). For rectal swabs, the viral RNA copies were significantly decreased from 4 dpi at both doses (Supplementary Fig. 5a). The reduction of viral loads in the upper and lower respiratory tract was consistent with an improvement in clinical symptoms and lung pathology (Supplementary Table 4 and Supplementary Fig. 6). To compare the therapeutic effect of CT-P59 with a US FDA approved drug, remdesivir was

administered daily for 5 days (18 mg/kg) in ferrets following 1 day of SARS-CoV-2 infection. Remdesivir-treated ferrets showed attenuated virus titers and viral RNAs in lungs and nasal wash, respectively, compared with those of isotype control-treated animals, but the infectious virus was detected in the nasal wash until 6 dpi (Figs. 3a, b and 4a, d) suggesting delayed clearance of SARS-CoV-2 in ferrets compared with the CT-P59-treated group.

In SARS-CoV-2-infected golden Syrian hamsters, all doses with 15, 30, 60, or 90 mg/kg of CT-P59 24 h after virus challenge reduced the levels of the viral RNA from lungs at 5 dpi (Fig. 4e).

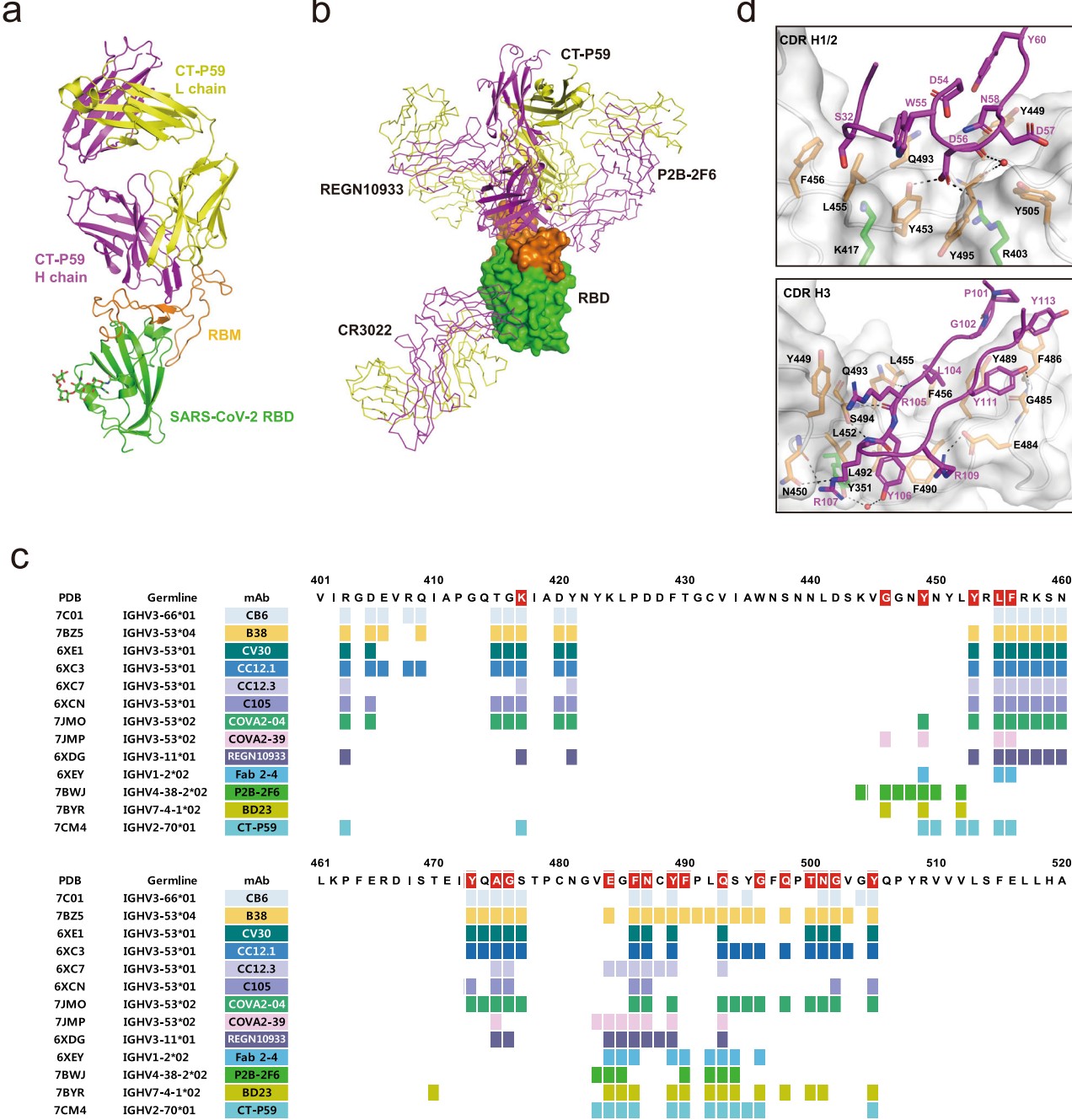

**Fig. 2 The structure of CT-P59 Fab in complex with SARS-CoV-2 RBD. a** The overall structure of the CT-P59 Fab/SARS-CoV-2 RBD complex. The RBD domain is green for the core subdomain, and orange for RBM. The heavy and light chains of CT-P59 are magenta and yellow, respectively. **b** Superposition of the neutralizing antibodies in complex with RBD. RBD is shown as a surface model. CT-P59 is shown as a cartoon, and the other antibodies (CR3022: PDB 6XC3, PB2-2F6: PDB 7BWJ, REGN10933: PDB 6XDG) are shown as a ribbon model. The heavy and light chains of Fab are magenta and yellow, respectively. **c** Assignment of the epitope residues for RBD-targeting neutralizing antibodies with a distance cutoff of 4.5 Å. RBD residues interacting with ACE2 are highlighted in red. **d** The detailed interactions between the RBD and CDR loops of CT-P59. The interfaces between RBD and CDR H1/2 or H3 are shown in the top and bottom panels, respectively. The RBD domain is shown as a surface model with a semi-transparent representation. The CDR loops and interacting residues on the interfaces are shown in ribbons and sticks, respectively. The residues are colored as in **a**. Dashed lines indicate hydrogen bonds. Water molecules are shown as red spheres.

The viral load reached peak levels on 3 dpi (8.3 log $TCID_{50}$/g) in the lungs of vehicle control-treated hamsters and slightly declined by 5 dpi (6.8 log $TCID_{50}$/g). In CT-P59-treated hamsters, there was significant attenuation of viral loads in lungs in the 15 mg/kg treated group, and all other groups (30–90 mg/kg) showed no infectious viruses in lung tissues 48 h after CT-P59 treatment,

suggesting complete inhibition of SARS-CoV-2 replication in lungs of golden Syrian hamsters at a dose of 30 mg/kg (Fig. 4b).

In the rhesus monkey study, no apparent clinical manifestations, including fever, weight loss, and respiratory distress, were observed in both CT-P59- and vehicle control-treated animals. The viral load reached peak levels on 2 dpi (4.3 and 3.2 log

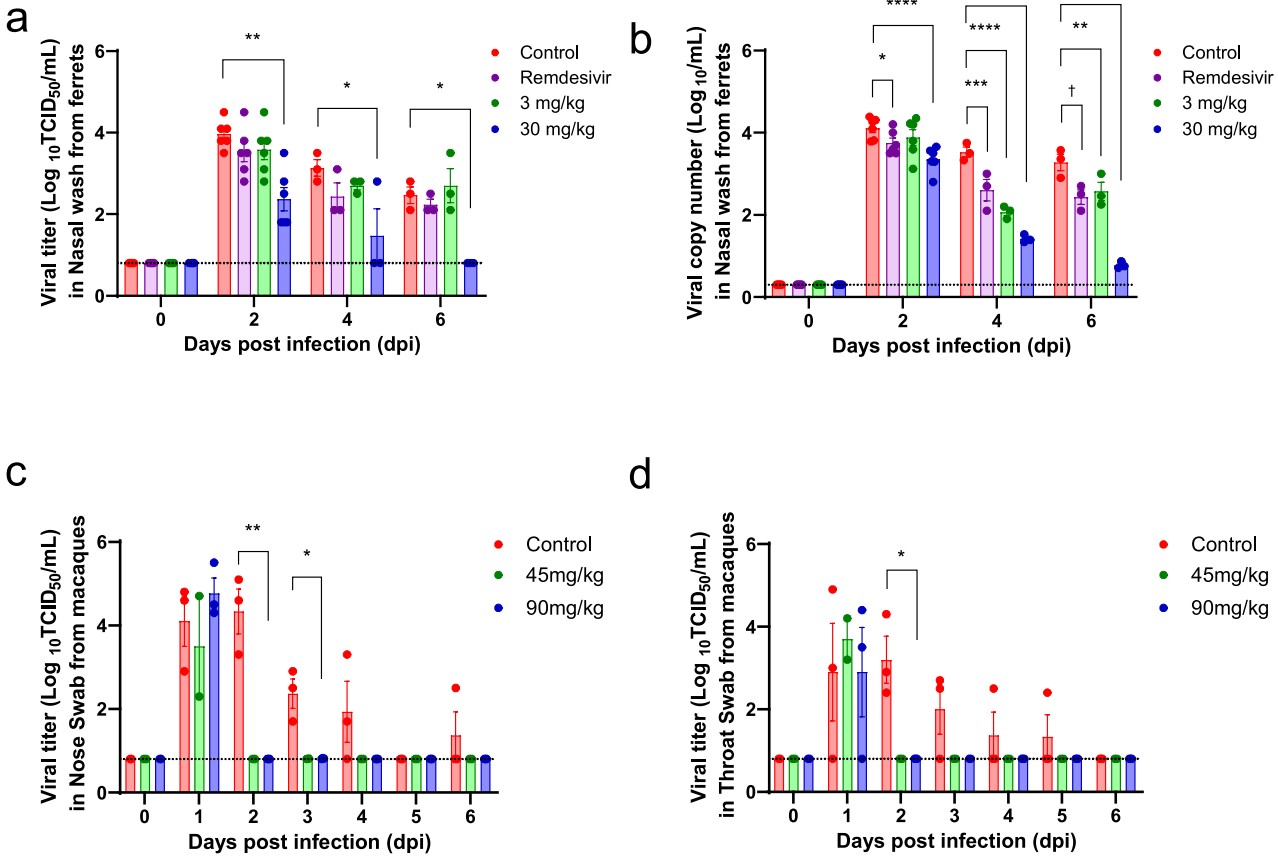

**Fig. 3 In vivo efficacy of CT-P59 in the upper respiratory tract in an animal model.** Female ferrets ($n = 6$/group), and rhesus monkeys (three control; two 45 mg/kg; three 90 mg/kg) were challenged with $10^{5.8}$ TCID$_{50}$/ml and $10^{6.4}$ TCID$_{50}$ of SARS-CoV-2, respectively. Control (ferrets: 30 mg/kg of human IgG isotype and rhesus monkeys: vehicle) and CT-P59 (ferrets: 3, and 30 mg/kg, rhesus monkeys: 45, and 90 mg/kg) were administered intravenously after 24 h of virus inoculation, respectively. To compare the efficacy of CT-P59, remdesivir (18 mg/kg per ferret) was administered daily via oral gavage after 24 h of virus inoculation in ferrets for 5 days. To detect viral load in the upper respiratory tract, nasal wash specimens were collected from ferrets at 2, 4, and 6 dpi, and nose and throat swab specimens from monkeys were collected daily up to 6 dpi. Virus titers (TCID$_{50}$) were measured in nasal wash/swab and throat swabs specimens from each group of (**a**) ferrets and, **c**, **d** rhesus monkeys. The viral RNA copy numbers were measured in nasal washes from (**b**) ferrets. Viral titers and RNA copy numbers are shown as mean values $+/-$ SEM and titers below the limit of detection are shown as 0.8 log$_{10}$TCID$_{50}$/ml or 0.3 log$_{10}$ viral RNA copies/ml (dashed lines). The asterisks and daggers indicate significance between the control and each group as determined by two-way ANOVA and subsequent Dunnett's test. *$P = 0.0001$ and **$P < 0.0001$ (**a**); *$P = 0.0456$, **$P = 0.0035$, ***$P = 0.0001$, ****$P < 0.0001$, and $^\dagger P = 0.0005$ (**b**); *$P = 0.0079$ and **$P < 0.0001$ (**c**); and *$P = 0.0021$ (**d**).

TCID$_{50}$/ml) in nasal and throat swabs, respectively, and then gradually declined until 6 dpi in vehicle control-treated group (Fig. 3c, d). In contrast, CT-P59 treatment rapidly reduced virus titers and the infectious virus was not detected in the upper respiratory tract even at 2 dpi following the CT-P59 administration in both 45 and 90 mg/kg groups. In addition, no viral RNAs were detected in rectal swabs collected from CT-P59-treated animals from 4 dpi (Supplementary Fig. 5b). All monkeys were euthanized at 6 dpi and individual lung lobes were collected to quantify the infectious virus titer. Although viral RNA persisted in middle and lower lobes, no infectious viruses were detected in any of the lung lobes tested from any animals, including vehicle control- and CT-P59-treated groups (Fig. 4c, f).

To further investigate the possible adverse effects, we performed the in vitro ADE assay with authentic SARS-CoV-2. The viruses infected permissive cells (VeroE6) and two Fc receptor-bearing cells; Raji cells (FcγRII expression) and U937 cells (FcγRI & II expression), followed by virus titration with an anti-nucleocapsid antibody. No increase in the viral infections was observed in Fc receptor-bearing cells; Raji and U937 (Fig. 5b, c) as well as VeroE6 (Fig. 5a).

## Discussion

In this study, we demonstrated the potential therapeutic benefit of neutralizing antibody CT-P59 targeting RBD of SARS-CoV-2 in vitro and in vivo studies. We found that CT-P59 binds to RBD of S protein, rendering complete steric hindrance interfering with the viral binding to ACE2 by BLI competition assay and X-ray crystallography. Importantly, CT-P59 significantly inhibited the viral replication of clinical isolates, wild-type, and D614G variant by in vitro PRNT.

SARS-CoV-2 RBD mutations might alter the binding affinity of the virus for ACE2[9,12,13]. For instance, V367F, W436R, and D364Y were reported to increase the binding affinity for ACE2, which might accelerate viral spread further perpetuating the pandemic. We found that CT-P59 binds to RBD mutant proteins and also interferes with ACE2 (Fig. 1b and Supplementary Table 1). In addition, according to the X-ray crystallography data

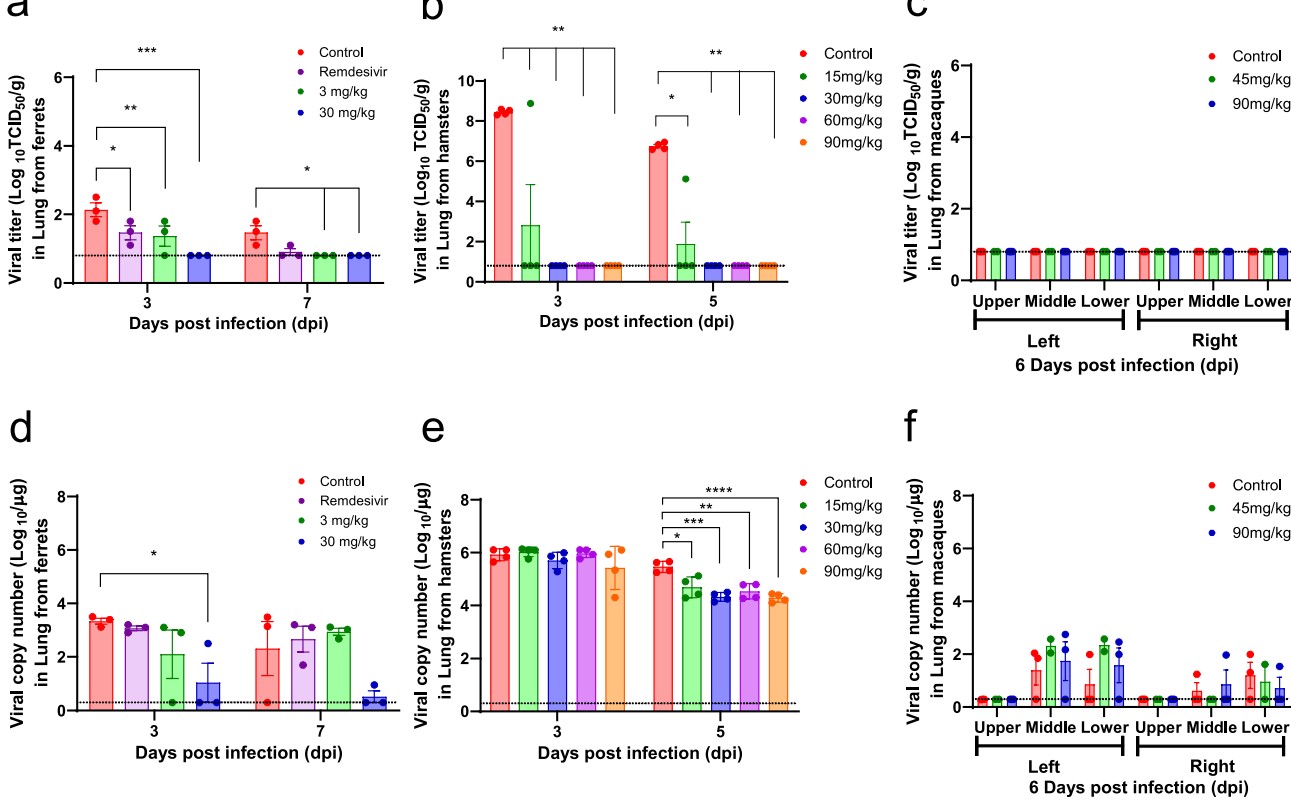

**Fig. 4 Virus titration and quantitation in the lower respiratory tract of animal models.** Three ferrets per group were euthanized at 3 and 7 dpi, and lungs were collected to measure viral titers (**a**) and the number of viral RNA copies (**d**). Golden Syrian hamsters ($n = 12$/group) were challenged intranasally with $6.4 \times 10^4$ PFU/80 μL of SARS-CoV-2. Vehicle and 15, 30, 60, and 90 mg/kg of CT-P59 were intraperitoneally administered 24 h after virus inoculation. Four animals were euthanized for virus titration (**b**) and quantitation of viral RNA copies (**e**) from each group at 3 and 5 dpi. Rhesus monkeys (control $n = 3$, 45 mg/kg $n = 2$, 90 mg/kg $n = 3$) were infected with $10^{6.4}$ TCID$_{50}$/ml of SARS-CoV-2 via in a combination of intranasal (0.5 ml), intratracheal (4 ml), ocular (0.25 ml/eye), and oral (1 ml) routes. Vehicle, 45 mg/kg and 90 mg/kg of CT-P59 were administered intravenously after 24 h of virus infection. All rhesus monkeys were euthanized at 6 dpi, and lungs were collected to measure viral titers (**c**) and the number of viral RNA copies (**f**). Viral titers in the lung were determined by TCID$_{50}$ assessment in Vero cells and viral RNA copy number measurement using qRT-PCR. Viral titers and RNA copy numbers are shown as mean value $+/-$ SEM and titers below the limit of detection are shown as 0.8 log$_{10}$TCID$_{50}$/ml or 0.3 log$_{10}$ viral RNA copies/ml (dashed lines). Asterisks indicate statistical significance between the control and each group as determined by two-way ANOVA and subsequent Dunnett's test. *$P = 0.0309$, **$P = 0.0131$, and ***$P < 0.0001$ (**a**); *$P = 0.0002$ and **$P < 0.0001$ (**b**); *$P = 0.0329$ (**d**); and *$P = 0.0123$, **$P = 0.0025$, ***$P = 0.0003$, and ****$P = 0.0002$ (**e**).

(Fig. 2c and Supplementary Fig. 4b), CT-P59 does not bind to the amino acid residues at position 367, 436, or 364 of the RBD. These results suggest that CT-P59 might be able to neutralize naturally occurring potential variants.

The complex structure of CT-P59 shows that CT-P59 inhibits SARS-CoV-2 RBD binding to its cellular receptor, ACE2, by blocking substantial areas of the ACE2 interaction regions. Among the previously reported neutralizing antibodies against SARS-CoV-2 RBD that specifically blocks ACE2 binding, we compared the publicly available atomic coordinates with those of CT-P59 to evaluate the association mode between antibodies and RBD (Supplementary Fig. 3a). We found that the majority of the ACE2 blocking antibodies—including CB6[16], B38[17], CV30[18], CC 12.1[19], CC 12.3[19], C105[22], COVA2-04[23], and REGN10933[14]—adopt a similar orientation when bound to RBD. Each of these antibodies belongs to the immunoglobulin heavy-chain variable region genes (IGHV) 3 germline that is the most frequently used IGHV gene among the known SARS-CoV-2- neutralizing antibodies[28]. The neutralizing antibody P2B-2F6[15] which is based on the IGHV4-38-2 gene, on the other hand, interacts with RBD at about a 90° angle from the previous group. Notably, CT-P59 (based on IGHV2-70) binds with an orientation in the middle of

these mAb groups (Supplementary Fig. 3a) and shares portions of the epitope from each group (Fig. 2c). To our knowledge, CT-P59 is the first SARS-CoV-2 RBD-neutralizing antibody with an IGHV2 germline lineage that its high-resolution structure reported. COVA2-39, another IGHV3 germline antibody, adopts a similar RBD-binding angle with CT-P59 that is quite different from the majority of other IGHV3 antibody groups (Supplementary Fig. 3a). Despite similar binding orientation to RBD in general, two antibodies face to different direction resulting in a distinct subset of epitope residues (Fig. 2c and Supplementary Fig. 3b). Cryo-electron microscopy has revealed that RBD of SARS-CoV-2 S protein trimer undergoes either "up" or "down" conformations and ACE2 can only bind to the "up" conformation[29,30]. Structural alignment of CT-P59 with SARS-CoV-2 S protein trimers showed that CT-P59 binds to RBD on the "up" conformation without any steric hindrance whereas it collides with the Asn343 glycosylated site on adjacent protomer in the "down" form (Supplementary Fig. 3c).

Because no animal models are available that accurately reflect clinical symptoms (e.g., lung damage) of patients with severe COVID-19[31–35], ferrets, Syrian hamsters, and rhesus monkeys have been used together for evaluation of SARS-COV-2

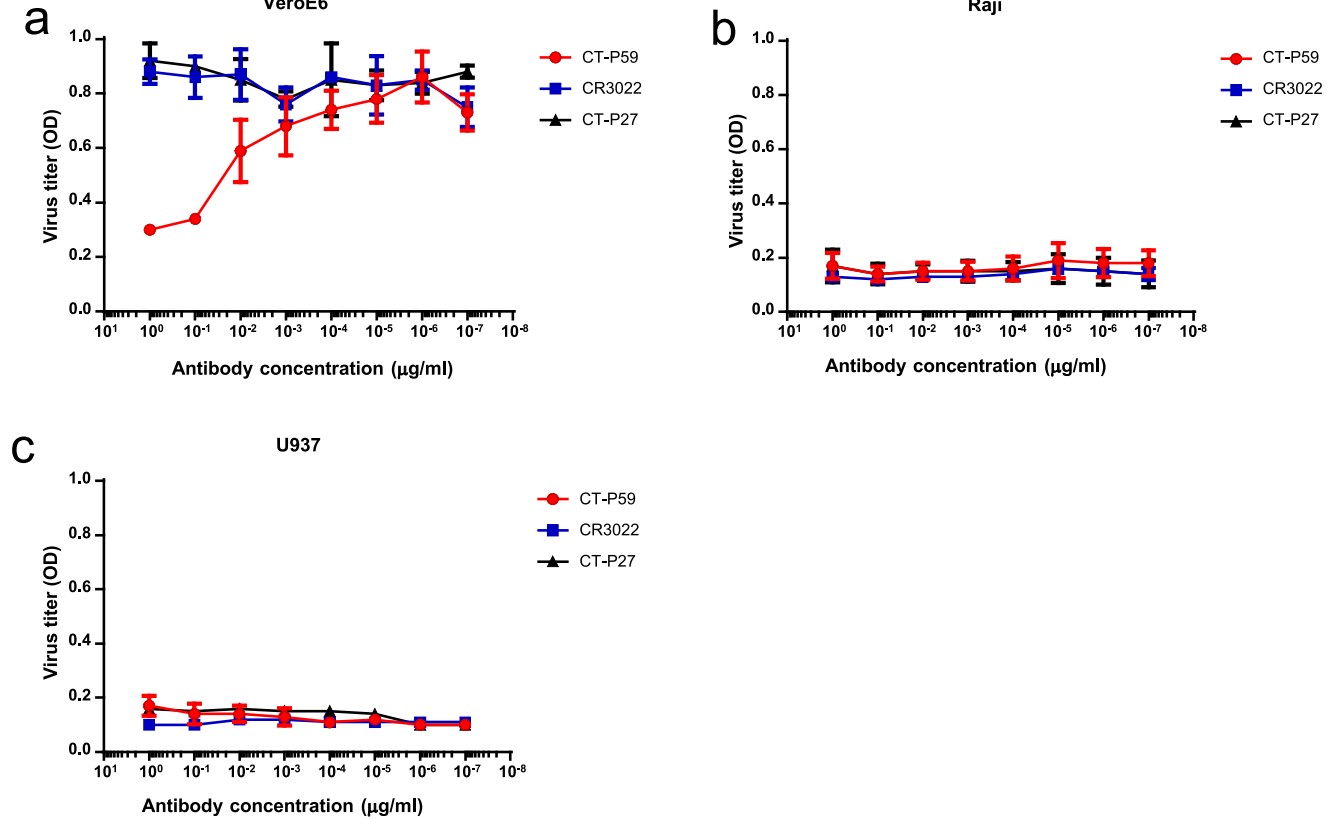

**Fig. 5 No evidence of ADE caused by CT-P59 bound to SARS-CoV-2 in permissive cells and Fc receptor (FcR)-bearing cells. a** FcR-independent ADE. SARS-CoV-2 was mixed with a wide range of antibodies; CT-P59 (red circle), CR3022 (blue square), and CT-P27 (black triangle). VeroE6 was infected with a virus–antibody complex. The virus titers were determined by optical density (OD) using an anti-nucleocapsid antibody. **b** FcγR II-dependent ADE. In vitro ADE assay was performed as described in (**a**) except Raji cells. **c** FcγR I&II-dependent ADE. In vitro ADE assay was carried out as described in (**a**), except U937 cells. The experiments were performed in triplicates (CR3022 and CT-P27) or in quadruplicates (CT-P59). Average and standard deviation of virus titers are depicted as dot and error bar, respectively.

pathogenesis/transmission and to assess the efficacy of therapeutics and vaccines against COVID-19[36–39]. In vivo challenge studies using these models have demonstrated that CT-P59 is capable of quickly decreasing virus titer (Figs. 3 and 4), particularly improving clinical symptoms and pathological changes in ferrets (Supplementary Table 4 and Supplementary Fig. 6). Furthermore, CT-P59 exhibits a better neutralizing effect by measuring the TCID$_{50}$ assay in the upper and lower airways, which indicates that CT-P59 could prevent SARS-CoV-2 virus from replicating in vivo. Notably, when we compared the therapeutic efficacy of CT-P59 with remdesivir, a drug for use in hospitalized patients with COVID-19, the CT-P59-treated ferrets showed more attenuated viral loads in upper respiratory tracts from 2 dpi (Fig. 3a, b). The early clearance of infectious virus suggests that CT-P59 might be an option for COVID-19 patients as a combination therapy.

Concerning ADE[40,41], the in vitro assay indicated no CT-P59-mediated increase in authentic viral infections in FcR-bearing cells and permissive cells (Fig. 5), in line with no worsening of symptoms in CT-P59-treated animals as described above. Moreover, a recent animal study showed that ADE was not observed by vaccine-targeting SARS-CoV-2 RBD[11]. Therefore, these observations suggest that CT-P59 can neutralize SARS-CoV-2 via binding to RBD and ameliorate pathological symptoms without ADE during clinical trials.

In summary, we successfully identified the RBD-specific mAb and characterized the structural mode of action and antiviral effect of the novel SARS-CoV-2 neutralizing antibody via in vitro studies, thereby substantiating in vivo efficacy in three animal models. Thus, CT-P59 is a promising treatment for COVID-19 patients as well as a prophylactic option. The effectiveness and safety of CT-P59 are proved in phase I clinical trial and currently underway in phase II clinical trial in South Korea and other countries.

## Methods

**Cells and viruses**. VeroE6 cells (ATCC, CRL-1586) were cultured in Dulbecco's modified Eagle's medium (DMEM) supplemented with 5% (v/v) fetal bovine serum (FBS) and penicillin–streptomycin (100 U/ml). The SARS-CoV-2 viruses used for in vitro PRNT assay were propagated in VeroE6 cells with DMEM supplemented with 2% FBS[42]. BetaCoV/Korea/KCDC03/2020 (Accession ID: EPI_ISL_407193) and hCoV-19/South Korea/KUMC17/2020 (provided by microbiology lab in Korea University) were isolated from Korean COVID-19 patients. The SARS-CoV-2 virus (NMC-nCoV02, isolated from a Korean COVID-19 patient) used for TCID$_{50}$ and ferret challenges was propagated in Vero cells (ATCC, CCL-81). Raji cells (ATCC, CCL-86) and U937 cells (ATCC, CRL-1593.2) were cultured with RPMI-1640 containing 10% FBS and PenStrep (Gibco). Authentic virus infection and animal challenges were conducted in biosafety level-3.

**Isolation of PBMCs from COVID-19 patient**. Blood was collected from a convalescent COVID-19 patient in Korea with approval by Seoul National University Hospital Institutional Review Board (IRB No. 2002-105-110). Samples were obtained 48 h after the disappearance of symptoms, and two consecutive respiratory specimens at an interval of 24 h were confirmed as negative for SARS-CoV-2 by PCR before blood sampling. Peripheral blood mononuclear cells (PBMCs) were isolated from the collected blood using Ficoll-Paque (GE Healthcare), and mRNA was extracted using the TRIzol reagent (Thermo Fisher). The isolated mRNA was immediately converted to cDNA using SuperScript$^{TM}$ III Reverse Transcriptase (Invitrogen).

**Phage library construction and biopanning**. Antibody variable regions ($V_L$ and $V_H$) were amplified by PCR with appropriate primers for phage display. Single-chain variable fragments (scFvs) were generated by linking $V_L$ and $V_H$ fragments and directly cloned into phagemid vector, pComb3xSS, for library construction. ER2738 cells (Lucigen) were transformed with the scFv library, then cultured in SB medium containing 50 μg/ml carbenicillin and VCSM13 helper phage (Stratagene) at 37°C overnight. Next day, phages displaying scFv were harvested for biopanning to screen SARS-CoV-2 RBD-binding scFv displayed on phage. Briefly, SARS-CoV-2 RBD (Sino biological) was coated on magnetic beads (Invitrogen) and incubated with the phage library. Following incubation and washing, SARS-CoV-2 RBD-bound phages were eluted and used to infect fresh ER2738 cells. After several rounds of biopanning, scFv phages binding to SARS-CoV-2 RBD were identified by phage ELISA for further selection.

**Preparation of scFv-Fc, full-length IgG, and S proteins**. Each scFv identified by phage ELISA was cloned into the Fc fusion vector and transiently expressed in Chinese hamster ovary (CHO) cells. Next, for the expression of full-length IgG, synthesized DNAs of heavy chain and light chain for each mAb were inserted into MarEx vectors (Celltrion) by enzymatic digestion with *Nhe*I (NEB)/*Pme*I (NEB) and *Hpa*I (NEB)/*Cla*I (NEB), respectively. CR3022 antibody was reconstituted with variable sequences for light and heavy chain according to the published sequence information (US8,106,170B2). Thereafter, transient expression by co-transfection was performed in CHO cells. Each scFv-Fc and full-length IgG was purified with affinity chromatography on Protein A (GE Healthcare). For the production of SARS-CoV-2 RBD, DNA encoding the SARS-CoV-2 S protein RBD (YP_009724390.1: Arg319-Asn536) with a polyhistidine tag at the C-terminus was cloned into the MarEx vector and transiently expressed in CHO cells. SARS-CoV-2 RBD with polyhistidine tag was affinity purified using Ni-NTA Resin (Thermo Fisher). Recombinant proteins for RBD and its mutants (A435S, F342L, G476S, K458R, N354D, V367F, V483A, W436R), SARS-CoV S1, HCoV-HKU1 S1, and MERS-CoV RBD were commercial products (Sino Biological).

**mAb-neutralizing assays**. To evaluate the neutralizing activity of monoclonal antibodies, plaque reduction neutralizing tests for SARS-CoV-2 were performed as described previously[43]. Briefly, twofold serially diluted mAbs ranging from $10^3$ to 1 ng/ml and an equal volume of virus (40 pfu/well) were incubated at 37 °C for 2 h. The antibody–virus mixture was inoculated into a 24-well plate seeded with VeroE6 cells ($1 \times 10^5$ cells/well) and incubated at 37 °C for 1 h, followed by an overlay of 1 ml of 0.5% agarose (Lonza). After 2 to 3 days of incubation, the cells were fixed with 4% paraformaldehyde and visualized plaques with crystal violet. Two independent experiments were performed in duplicate for each mAb. The data were fitted to a dose–response inhibition model, and the half-maximal inhibitory concentration ($IC_{50}$) of each mAb was calculated using GraphPad Prism6 software.

**Surface plasmon resonance for affinity**. The binding affinity of CT-P59 to SARS-CoV-2 RBD was assayed using a Biacore T200™ SPR instrument (Cytiva). SARS-CoV-2 RBD manufactured by Celltrion was covalently immobilized on the CM5 chip using an amine coupling reaction. Any unbound, SARS-CoV-2 RBD was removed by at least six washes of the pre-run solution before the sample run. CT-P59 was serially diluted from 10 to 0.04 nM using HBS-EP buffer (pH 7.4) and then injected for 120 s, followed by HBS-EP buffer (pH 7.4) for 120 s to generate the binding and dissociation curves, respectively. After each cycle, the chip surface was treated with a brief pulse of 20 mM NaOH until the response unit (RU) signal returned to baseline, and then a new cycle was started. The dissociation constant was fitted to a bivalent analyte model using Biacore evaluation software (Cytiva).

**Biolayer interferometry (BLI)**. Competitive binding to SARS-CoV-2 RBDs between CT-P59 and ACE2, binding specificity, and binding affinity to SARS-CoV-2 RBDs were measured by BLI using the Octet QK$^e$ system (ForteBio). All samples were prepared with corresponding concentration by dilution in Kinetic Buffer (ForteBio). To determine the competitive characteristics between CT-P59 and ACE2, the immobilized wild-type and mutant SARS-CoV-2 RBD proteins with a concentration of 50 nM were saturated with 267 nM of CT-P59 for 5 min, and then flowed with CT-P59 (133.5 nM) in the presence or absence of ACE2 (133.5 nM) for 5 min. As a positive control, the buffer was loaded onto SARS-CoV-2 RBD immobilized biosensor and flowed ACE2 (133.5 nM). For the binding specificity assay, binding of CT-P59 to four virus S proteins (SARS-CoV-2 RBD, SARS-CoV S1, HCoV-HKU1 S1, MERS RBD) was measured. Each S protein (50 nM) was loaded onto Anti-Penta-HIS Biosensor, and then CT-P59 (267 nM) flowed for 5 min. To evaluate binding affinity between CT-P59 and SARS-CoV-2 RBD wild type and mutants, CT-P59 (5 nM) was loaded onto anti-human IgG Fc Capture Biosensor (Sartorius) for 7.5 min, and then each SARS-CoV-2 RBD was flowed with the concentration of 0 nM, 2.5 nM, 5 nM, 10 nM, and 20 nM for 10 min and 15 min to generate association and dissociation curve, respectively. BLI data were collected using Octet Data Acquisition v11.0 software (ForteBio), and data analysis was performed with ForteBio Data Analysis v11.0 and Data Analysis HT v11.0 software (ForteBio).

**Crystallization and structure determination**. The CT-P59 Fab/SARS-CoV-2 RBD complex was prepared by mixing the purified SARS-CoV-2 RBD protein with CT-P59 Fab at a 1:1.2 molar ratio. Excess CT-P59 Fab was removed by size-exclusion chromatography by using a 16/600 Superdex-200 column (GE Healthcare) equilibrated in 10 mM Tris-HCl (pH 8.0) and 150 mM NaCl. The fractions containing the complex were pooled and concentrated up to 6 mg/ml and used for crystallization experiments. Diffraction quality crystals were obtained at 20 °C by the sitting-drop vapor-diffusion method with 0.4 μl protein solution mixed with 0.4 μl of the precipitant solution containing 10 mM $NiCl_2$, 0.1 M Tris-HCl (pH 8.0) and 16% (wt/vol) PEG MME 2000. For data collection, the crystals were cryo-protected by briefly soaking in the precipitant containing an additional 20% ethylene glycol and immediately mounted in a stream of gaseous nitrogen at 100 K. X-ray diffraction data were collected at a wavelength of 0.9796 Å using a beamline BL-5C of Pohang Light Source-II, Republic of Korea. The dataset was processed using XDS program package[44]. The CT-P59 Fab/SARS-CoV-2 RBD complex structure was determined by molecular replacement using Phaser[45]. The SARS-CoV-2 RBD/CB6 complex structure (PDB code, 7C01) was used as a search model. Model building and refinement were performed using Coot[46] and the Phenix package[47], respectively. The Ramachandran statistics of the final structure are 96.81% in the most favored region, 3.19% in the allowed region, and 0.00% in the disallowed region. The X-ray diffraction and structure refinement statistics are summarized in Supplementary Table 2. All structure figures were generated with PyMol[48]. The solvent-accessible surface area was calculated using the online PISA service (http://www.ebi.ac.uk/pdbe/prot_int/pristart.html)[26].

**In vitro ADE assay**. VeroE6 cells ($1 \times 10^4$/well) were seeded in a 96-well plate at 24 h before infection. Next day, antibodies (CT-P59, CR3022, and CT-P27) were tenfold serially diluted from 2 to $2 \times 10^{-7}$ μg/ml with serum-free media, OptiPRO SFM containing L-Glutamine (Gibco). Equal volume of SARS-CoV-2 (BetaCoV/Korea/KCDC03/2020) viruses (0.05 moi) was mixed with the diluted antibodies in the 96-well plate block (Corning) for 2 h at 37 °C $CO_2$ incubator. In parallel, Raji and U937 cells ($2 \times 10^4$/well) were prepared in U-bottom 96-well plate. The inoculum from the plate block was added into VeroE6 cells, Raji cells, and U937 cells and then incubated for 24 h at 37 °C $CO_2$ incubator. At 24 h post-infection, the cells were fixed with 80% acetone (Sigma). After washing, the cells were probed with SARS-CoV-2 anti-nucleocapsid antibody (Sino Biologicals, 1:2000), and bound antibody was detected by horseradish peroxidase (HRP)-conjugated anti-mouse antibody (Southern Biotech, 1:4000). Tetramethylbenzidine (TMB) was added and incubated for 5 min, then stopped by $H_2SO_4$. Virus titers were assessed by optical density measured by spectrophotometer (Thermo Scientific) at 450 nm.

**Animal experiments**. Ferret, golden Syrian hamster, and rhesus monkey studies were carried out according to the procedures approved by the Institutional Animal Care and Use Committee of Chungbuk National University (CBNUA-1352-20-02), Agency for Defense Development (ADD-IACUC-20-12), and Korea Research Institute of Bioscience and Biotechnology (KRIBB-AEC-20168), respectively, and complied with all relevant ethical regulations regarding animal research. In in vivo experiments, specific pathogen-free (SPF) animals were used in the Animal Biosafety Level 3 (ABSL-3) facilities for experiments with infectious virus.

**Ferret study**. Groups of 14- to 18-month-old female ferrets (Mustela putorius furo, $n = 6$/group) were sourced from ID.BIO, Corp., Korea. All ferrets which were seronegative for SARS-CoV-1 and SARS-CoV-2 were inoculated intranasally and intratracheally with $10^{5.8}$ $TCID_{50}$ of NMC-nCoV02 (total 1 ml) under anesthesia. Two doses, 3 and 30 mg/kg, of CT-P59 were administered intravenously 24 h after virus inoculation in each group. Animals in the control group were given 30 mg/kg of human IgG isotype. Remdesivir (18 mg/kg) was administrated daily via oral gavage 24 h post inoculation for 5 days. Body weights and temperatures were measured, and nasal washes and rectal swab specimens were collected every other day. Three ferrets per group were euthanized at 3 and 7 dpi, and the lungs were subjected to measure tissue viral titers and examine histopathology. Animals were euthanized by the administration of potassium chloride solution following anesthetized condition by administration of Xylazine (0.15 ml/kg, intramuscular route) and Alfaxan (0.5 ml/kg, intravenous route). Viral titers in nasal washes and lungs were determined by $TCID_{50}$ assay in Vero cells, while the viral loads in nasal washes, lungs, and rectal swab specimens were assessed using quantitative real-time PCR (qRT-PCR).

**Golden Syrian hamster study**. Five-week-old male golden Syrian hamsters (Mesocricetus auratus, $n = 12$/group) sourced from Janvier Labs, France, were challenged with $6.4 \times 10^4$ PFU/80 μl of SARS-CoV-2 via an intranasal route. Vehicle and 15, 30, 60, and 90 mg/kg CT-P59 were administered via an intraperitoneal route 24 h after virus inoculation. At 3 and 5 dpi, four animals from each group were euthanized for quantification of viral load in the lungs. Animals were euthanized by deep anesthesia through the isoflurane inhalation.

**Rhesus monkey study**. Eight 5- to 7-year-old rhesus monkeys (Macaca mulatta, five males, three females) were sourced from three suppliers in China, including Suzhou Xishan Zhongke Laboratory Animal co., Ltd, Biomedical Research. co., Ltd,

and Guangzhou Monkey king Biotechnology co., Ltd. All monkeys were challenged with $2.6 \times 10^6$ TCID$_{50}$ of SARS-CoV-2 via a combination of intranasal (0.5 ml), intratracheal (4 ml), ocular (0.25 ml/eye), and oral (5 ml) routes. CT-P59 (45 mg/kg (one male, one female) and 90 mg/kg (two males, one female)) was administered intravenously 24 h after virus inoculation. Animals in the control group (two male, one female) were given an equal volume of vehicle. Viral load was measured by nasal, throat and rectal swabs daily until 6 dpi, and viral load in the lung was measured by necropsy at 6 dpi. Animals were euthanized after an intravenous overdose of T61 (0.3 mg/kg).

**Virus titration and quantitation**. Virus titers in nasal washes and lungs (TCID$_{50}$) and in rectal swab (qRT-PCR) were determined. Briefly, viral titers for samples from the upper and lower airway were measured using Vero cells. All tissue samples were diluted tenfold (w/v) with the sterile phosphate-buffered solution (PBS, pH 7.4) and homogenized using a tissue homogenizer (Precellys Homogenizer, Bertin Instruments, France). After centrifugation of all swabs and homogenized tissue samples at $3000 \times g$ for 10 min, the supernatants were filtered through a 0.2-μm pore-size syringe filter (Millipore, USA) and directly inoculated into Vero cells. The cells were incubated for 3 days at 37 °C and then stained with crystal violet for cytopathic effect (CPE). The values of TCID$_{50}$/ml were determined using a Reed and Muench method.

In the ferret and hamster studies, viral RNA was extracted using RNeasy kit (Qiagen) and cDNAs were reverse transcribed with a SARS-CoV-2-specific primer using QuantiTect Reverse Transcription (Qiagen). qRT-PCR reactions were performed using a SYBR Green Supermix (Bio-Rad). The viral RNA copy numbers were assessed by a CFX96 Touch Real-Time PCR Detection System (Bio-Rad). In the rhesus monkey study, total RNAs were accessed by RT-qPCR. Viral RNA was extracted using a commercial viral RNA extraction kit (QIAamp Viral RNA Mini Kit, Qiagen). RT-qPCR was performed with an S gene-based-SARS-CoV-2-specific primer according to a previous report (Supplementary Table 5)[36]. SARS-CoV-2 RNA standard samples were run in parallel for the determination of virus copy number in all reactions. Virus titer and viral RNA from animals were analyzed using GraphPad Prism v8.2.0.

**Histology**. Lung histology is evaluated as follows. Sections of the left caudal lung lobes were microscopically observed. Before collection, the lung lobes (with trachea intact) were insufflated with 10% neutral buffered formalin (NBF) and then submerged in 10% NBF for 2–3 days. Following fixation, the desired sections of lungs were embedded in paraffin, sectioned (5 μm), placed on glass slides, and stained with hematoxylin and eosin (H&E).

**Human samples**. The human samples were obtained according to the procedures approved by Seoul National University Hospital Institutional Review Board and complied with all relevant ethical regulations regarding human research. The blood was taken from a convalescent COVID-19 patient after she/he signed the informed consent form.

**Reporting summary**. Further information on research design is available in the Nature Research Reporting Summary linked to this article.

## Data availability

The data that support the findings of this study are available from the corresponding authors upon reasonable request. The atomic coordinates and structure factor files for the CT-P59 Fab/SARS-CoV-2 RBD complex have been deposited in the Protein Data Bank (PDB) under accession number 7CM4. The publicly available PDB codes used for the structural comparison are 6LZG, 7C01, 7BZ5, 6XE1, 6XC3, 6XC7, 6XCN, 7JMO, 7JMP, 6XDG, 6ZCZ, 7CAH, 6XEY, 7BWJ, 7BYR, and 6VXX. The accession IDs for viruses are YP_009724390.1 and EPI_ISL_407193. The sequence information for CR3022 reconstitution is US8,106,170B2. Source data are provided with this paper.

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

## Acknowledgements

This study was supported by a Korea National Institute of Health fund (2020-ER5311-00, 2020-ER5323-00, 2019-NI-077-01, 2019-NG-044-01), grants (011555-012664201) from the Agency for Defense Development, grants (PRM1752011) from the Ministry of Science and ICT, and partially supported by the Korea Research Institute of Bioscience and Biotechnology (KRIBB) Research Initiative Program (KGM9942011 and KGM4572013), Republic of Korea.

## Author contributions

C.K. and J.M.S. initiated and managed to identify CT-P59 from a convalescent patient with COVID-19. Y.J.A. narrowed down RBD-binding candidates by ELISA with help of C.K. and J.M.S. M.S.K. and G.S.P. carried out cloning and expression of CT-P59 and recombinant RBD protein. M.K. and H.N. conducted BLI with Octet. J.S.P. and M.S.L conducted SPR. E.Y.S. and J.N.L. carried out purification of antibody and antigen. P.K. contributed to the mode-of-action study with D.K.R. who performed in vitro ADE. J.I.K. and D.S. managed and analyzed animal experiments. K.S.K. and S.Y.L. coordinated the overall CT-P59 project. J.Y.L., H.L., J.S.Y., K.C.K., S.S.K., H.M.W., and J.W.K. isolated human PBMC from whole blood and performed in vitro PRNT with the help of C.K., J.M.S., and M.K. against wild and D614G variant with the help of M.S.P. Y.K.C., Y.I.K., K.M.Y., S.M.K., E.H.K., and S.J.P. performed animal experiments with ferrets. S.T.J., C.H.Y., Y.S., and S.H.G. performed animal experiments with golden Syrian hamsters with the help of J.Y.L., J.W.K., H.M.W., J.S.Y., and K.C.K. who performed in vitro TCID$_{50}$. H.O., B.S.K., J.J.H., and C.M.R. performed animal experiments with rhesus monkeys. W.B.P. and M.D.O. recruited patients recovered from COVID-19. Y.G.K., J.H.J., J.L., and J.S.B. carried out crystallization, diffraction data collection, structure determination, and analysis of CT-P59 and RBD complex. C.K., D.K.R., J.L., J.I.K., Y.K.C., and S.Y.L. analyzed the data and wrote the paper.

## Competing interests

Patents have been filed for CT-P59. C.K., D.K.R., J.L., J.M.S., M.K., J.I.K., P.K., J.S.B., E.Y.S., M.S.L., M.S.K., H.N., G.S.P., J.S.P., D.S., Y.A., J.N.L., K.S.K., and S.Y.L. are employees of Celltrion, Inc. This work was funded by Celltrion, Inc. and several grants listed in Acknowledgements. The remaining authors declare no competing interests.
