## [Peer Review File · Nature Communications]

REVIEWER COMMENTS

Reviewer #1 (Structural biology, humoral response to viral infection) (Remarks to the Author):

This paper is one in a line of many papers reporting antibodies to SARS-CoV-2. Many of these antibodies have been reported to bind to the RBD with similar properties. Thus, the use of 'novel' in the title is not warranted and certainly not without a lot more compelling analyses. 'Novel' should not generally be used in any case.

Page 5, lines 95-98. "Intriguingly, we found that CT-P59 more effectively reduced the replication of D614G variant with the value of IC50 (5.73 ng/ml) than that of the wild virus having D614 amino acid in S protein (Fig. 1a)." While this is interesting, the difference from wild-type is not that different and not clear if meaningful. It could also result from different numbers of spikes on the virus surface in WT and D614G. Please cite this actual WT value here in the text. I also don't think that the IC50 value is really meaningful to 2 decimal points.

Page 5, line 104. "CT-P59 has high affinity for SARS-CoV-2 RBD " In this paragraph, both IgG and single-chain Fv are mentioned. Are these binding studies done with IgG or and are avidity effects of the IgG likely to play a role in this assay?

Page 5, lines 104-105. "KD value of 27 pM" I don't see a value of 27pM. I see a value of 251pM in this table but a value of 27pM in Extended Fig. 1. Why are their 10- fold differences in the two reported measurements?

Page 5, line 106. Please replace "(SARS, HCoV-HKU1, and MERS)" with "(SARS-CoV, HCoV-HKU1, and MERS)

Page 5, line 107. Please replace "can bind" with "binds".

Page 6, line 114. "report" Can the authors compare with some of the other SARS-CoV-2 antibodies with structures available, and make sure they do not use a similar association angle? Some other antibodies with structures available are: H014 (7CAH); BD23 (7BYR); EY6A (6ZCZ); 2-4 (6XEY); C105 (6XCN); COVA2-04 (7JMO); COVA2-39 (7JMP).

Page 6, line 114. Please replace "(Fig. 2b)" with "(Fig. 2b and Extended Data Fig. 4a)".

Page 6, line 115. Please replace "are unique" with "is unique". However, until an analysis of all available antibodies structures is reported the epitope cannot be defined as unique.

Page 6, line 117. "solvent-accessible surface" Can the authors please describe or reference the method used for the calculation?

Page 6, line 117. "of 824.6 and 112.5 Å²" Decimal point are not meaningful- please truncate to integers.

Page 6, lines 121-122. "the β-hairpin structure of the CDR H3" What is the length of CDR H3?

Page 7, lines 147-148. "At 2 dpi, the infectious virus titre (TCID50) in nasal wash was significantly decreased when compared to controls in time- and dose-dependent manners" It is not clear what is being compared here - all samples or only CPT-P59 at 30mg/kg. The first P value is barely if not even significant.

Page 8, line 159. Please delete "the".

Page 8, line 174. "45 and 90 mg/kg" It would have been helpful to also have done lower amounts.

Page 9, lines 190-193. "Unlike RBD-targeting antibodies with similar sensitivities to the viral neutralization on a D614G variant²¹, CT-P59 can effectively neutralize D614G variant; the underlying mechanism remains to be elucidated in terms of S protein stability and viral kinetics" I don't really know what the sentence means. The D614G mutation is not in the RBD or RBM. It is good to know that there is no real difference in neutralization, but I am not sure what underlying mechanism is being referred to here.

Page 11, lines 227-229. "We therefore propose that CT-P59 may have more chance to block ACE2-RBD interaction in pre-fusion state than IGHV3 group antibodies if there is slight hinge-like movement in RBD region". It is not known what is meant by pre-fusion state here- it cannot be just all RBD down as cryo-electron tomography has shown up RBD up states also on viruses. All of the states alluded to here seem to be pre-fusion.

Page 11, line 239-240. "The early clearance of infectious virus suggests that CT-P59 might be an option for COVID-19 patients as combination therapy". How does CT-P59 compare with other antibodies that have reported animal studies?

Page 12, lines 245-247. "Therefore, these observations suggest that CT-P59 can remarkably neutralize SARS-CoV-2 via binding to RBD and ameliorate pathological symptoms without ADE during clinical trials". The lack of any ADE is important to show, but 'remarkably' is far too strong and should be removed.

Page 29, lines 584-585, Reference 43. Different font from the rest of the text.

Extended Data Table 2 | Data collection and refinement statistics
Please add number of unique reflections measured, Rpim, CC1/2, Wilson B.

α , β , γ ($^{\circ}$)

Please replace "90.00, 90.00, 90.00" with "90 90 90".

Rsym or Rmerge "0.245 (2.145)" 2 decimal points are sufficient and for high res >2.

I / σ I "12.44 (1.63)" 1 decimal point is sufficient.

Completeness (%) "99.78 (99.88)" 1 decimal point is sufficient.

Rwork / Rfree "0.2167 / 0.2420" 3 decimal points are sufficient.

B- values, please truncate to integers, decimal points are not meaningful.

Page 4, Extended Data Fig. 1, line 5. Please replace "Data is" with "Data are".

Reviewer #2 (Viral immunity) (Remarks to the Author):

Kim et al. report a therapeutic mAb against SARS-CoV-2 spike RBD. The study is comprehensive with results from virology, biophysics, structure, and animal efficacy. The manuscript is well written. This reviewer supports the publication in Nature Communications with some minor modifications.

1. For the in vivo efficacy results on respiratory samples, the authors used infectious virus amounts as an efficacy readout. This could be complicated with administered mAb that may inhibit the infectious virus during neutralization assay. So, it would be important to test one set of these samples using RT-PCR method.
2. The order of Extended Data figures should be reorganized in the order of their appearance in text.
3. Lines 104-105: the K_d value of 27 pM does not match " 2.51×10^{-10} M" in Extended Data Table 1.
4. Remove "novel" in the manuscript title.

Point-by-point responses to reviewers' comments

<Point-by-point Response to the reviewers' comments>

Reviewer #1 (Structural biology, humoral response to viral infection) (Remarks to the Author)

1. This paper is one in a line of many papers reporting antibodies to SARS-CoV-2. Many of these antibodies have been reported to bind to the RBD with similar properties. Thus, the use of 'novel' in the title is not warranted and certainly not without a lot more compelling analyses. 'Novel' should not generally be used in any case.

Response: We have replaced 'Novel' with 'Therapeutic' in the manuscript title

2. Page 5, lines 95-98. "Intriguingly, we found that CT-P59 more effectively reduced the replication of D614G variant with the value of IC₅₀ (5.73 ng/ml) than that of the wild virus having D614 amino acid in S protein (Fig. 1a)." While this is interesting, the difference from wild-type is not that different and not clear if meaningful. It could also result from different numbers of spikes on the virus surface in WT and D614G. Please cite this actual WT value here in the text. I also don't think that the IC₅₀ value is really meaningful to 2 decimal points.

Response: We appreciate the reviewer's comments. We have revised this part as followings

"We found that CT-P59 more effectively reduced the replication of D614G variant with the value of IC₅₀ (5.73 ng/ml) than that of the wild virus having D614 amino acid in S protein (Fig. 1a)."

→

"We found that CT-P59 reduced the replication of D614G variant with the value of IC₅₀ (5.7 ng/mL) to a similar extent as the wild type virus."

Also, we adjusted IC₅₀ values to 1 decimal point (WT: 8.37 ng/mL → 8.4 ng/mL, D614G: 5.73 ng/mL → 5.7 ng/mL).

3. Page 5, line 104. "CT-P59 has high affinity for SARS-CoV-2 RBD " In this paragraph, both IgG and single-chain Fv are mentioned. Are these binding studies done with IgG or and are avidity effects of the IgG likely to play a role in this assay?

Response: We determined the binding affinity with full IgG, but not single-chain Fv of CT-P59 by SPR. We did not check the avidity effect of CT-P59 IgG on the assay.

4. Page 5, lines 104-105. "K_D value of 27 pM" I don't see a value of 27pM. I see a value of 251pM in this table but a value of 27pM in Extended Fig. 1. Why are their 10- fold differences in the two reported measurements?

Response: The difference of K_D value (27 pM vs. 251 pM) is originated from both analysis method and binding scheme (binding order of RBD and CT-P59 antibody). The K_D values were determined in two different methods such as surface plasmon resonance (SPR) using Biacore T200 and biolayer interferometry (BLI) using Octet. The K_D value of 27 pM was obtained from SPR with RBD as ligand and CT-P59 as analyte. The K_D value of 251 pM was determined by BLI with CT-P59 as ligand and RBD as analyte. Although we thought that the binding order applied on SPR reflects the physiological condition, BLI was performed with the different binding scheme to compare binding affinity of various RBDs listed in Extended Data Table 1 (Supplementary Table 1 in revised manuscript).

For clarity, we have re-arranged/revised the paragraph as below;

“We found that CT-P59 can bind to these mutants and completely inhibit binding between ACE2 and RBD mutants (Extended Data Table 1 and Fig. 1b). Next, surface plasmon resonance analysis demonstrated that CT-P59 has high affinity for SARS-CoV-2 RBD with a K_D value of 27 pM (Extended Data Fig. 1). Furthermore, CT-P59 binding specificity to other coronaviruses (SARS, HCoV-HKU1, and MERS) was evaluated by BLI, indicating that CT-P59 can bind specifically to SARS-CoV-2. (Extended Data Fig. 2).”

→

“We found that CT-P59 can bind to these mutants and completely inhibit binding between ACE2 and RBD mutants by BLI (Fig. 1b and Supplementary Table 1). Furthermore, CT-P59 binding specificity to other coronaviruses (SARS-CoV, HCoV-HKU1, and MERS-CoV) was evaluated by BLI, indicating that CT-P59 can bind specifically to SARS-CoV-2 (Supplementary Fig. 1). Next, surface plasmon resonance analysis demonstrated that CT-P59 has high affinity for SARS-CoV-2 RBD with a K_D value of 27 pM (Supplementary Fig. 2).”

5. Page 5, line 106. Please replace "(SARS, HCoV-HKU1, and MERS)" with "(SARS-CoV, HCoV-HKU1, and MERS)

Response: ‘SARS’ was replaced with ‘SARS-CoV’ in manuscript according to the reviewer's comment. Also, we replaced ‘MERS’ with ‘MERS-CoV’. Please refer to Response 4.

6. Page 6, line 114. "report" Can the authors compare with some of the other SARS-CoV-2 antibodies with structures available, and make sure they do not use a similar association angle? Some other antibodies with structures available are: H014 (7CAH); BD23 (7BYR); EY6A (6ZCZ); 2-4 (6XEY); C105 (6XCN); COVA2-04 (7JMO); COVA2-39 (7JMP).

Response: According to the reviewer's comments, we analyzed the association angle of the other SARS-CoV-2 antibodies in complex with the RBD in Supplementary Fig. 3a. The association angle of CT-P59 is different from those of the others. We re-wrote the sentence as below.

“The association angle between CT-P59 and the RBD is different from that reported for other neutralizing antibodies in complex with the RBD (Fig. 2b)^{14,15}.”

→

“The association angle between CT-P59 and the RBD is different from that reported for other structures available neutralizing antibodies in complex with the RBD (Fig. 2b and Supplementary Fig. 3a)¹⁴⁻²⁵.”

References 20 – 25 were added for H014 (7CAH); BD23 (7BYR); EY6A (6ZCZ); 2-4 (6XEY); C105 (6XCN); COVA2-04 (7JMO); COVA2-39 (7JMP).

We have also included assignment of epitope residues for the additionally evaluated SARS-CoV-2 antibodies in Fig. 2c except for H104 (7CAH) and EY6A (6ZCZ) that are not ACE2 blocker.

Among the additionally evaluated antibodies, COVA2-39 seem to adopt similar RBD binding angle with CT-P59 as shown in Supplementary Fig. 3a. However, two antibodies face different directions resulting in distinct subset of epitope residue as shown in Fig. 3c. We have added a new image containing detail structure comparison between CT-P59 and COVA2-39 in complexed with RBD in Supplementary Fig. 3b. The description was added in discussion section (page 11, line 217) as below.

“COVA2-39, another IGHV 3 germline antibody, adopts similar RBD binding angle with CT-P59 that is quite different from majority of other IGHV3 antibody group (Supplementary Fig. 3a). Despite of similar binding orientation to RBD in general, two antibodies face to different direction resulting in distinct subset of epitope residues (Fig. 2c and Supplementary Fig. 3b).”

7. Page 6, line 114. Please replace “(Fig. 2b)” with “(Fig. 2b and Extended Data Fig. 4a)”.

Response: We replaced “(Fig 2b)” with “(Fig. 2b and Supplementary Fig. 3a)”. Please refer to Response 6.

8. Page 6, line 115. Please replace “are unique” with “is unique”. However, until an analysis of all available antibodies structures is reported the epitope cannot be defined as unique.

Response: We agree that it is difficult to confer the uniqueness of the epitope of CT-P59 without analyzing all antibodies being reported almost every week. Therefore, we prefer to use “distinct” instead of “unique” in the sentence as below.

“These observations indicate that the epitope of CT-P59 are unique from those of other antibodies¹⁴⁻¹⁹ (Fig. 2c).”

→

“These observations indicate that the epitope of CT-P59 are distinct from those of other antibodies (Fig. 2c).”

9. Page 6, line 117. "solvent-accessible surface" Can the authors please describe or reference the method used for the calculation?

Response: According to the reviewer's comments, we included the method used in the sentence as below.

"Further, the interactions of the RBD with the heavy and light chains of CT-P59 bury a solvent-accessible surface area of 824.6 and 112.5 Å², respectively."

→

"Further, the interactions of the RBD with the heavy and light chains of CT-P59 bury a solvent-accessible surface area of 825 and 113 Å², respectively, calculated by PISA²⁶."

We have also added the following sentence in the method section of "Crystallization and structure determination" (page 17, line 359).

"Solvent accessible surface area was calculated using online PISA service (http://www.ebi.ac.uk/pdbe/prot_int/pristart.html)²⁶."

10. Page 6, line 117. "of 824.6 and 112.5 Å²" Decimal point are not meaningful- please truncate to integers.

Response: According to the reviewer's comment, the decimal points were truncated to integers. Please refer to Response 9.

11. Page 6, lines 121-122. "the β-hairpin structure of the CDR H3" What is the length of CDR H3?

Response: The CDR H3 is consisted with a total 18 amino acid residues and the β-hairpin structure of CDR H3 has 9 amino acid residues. Among 9 residues, 7 amino acids are involving in binding to RBD. We have added the information of CDR H3 length in the sentence as below.

"Of note, the β-hairpin structure of the CDR H3 plays a crucial role in the strong association with the RBD, by forming eight hydrogen bonds as well as hydrophobic interactions involving several of aromatic residues in the middle of the ACE2 binding surface (Fig. 2d)."

→

"Of note, the β-hairpin structure of the CDR H3 of 18 amino acids length plays a crucial role in the strong association with the RBD, by forming eight hydrogen bonds as well as hydrophobic interactions involving several of aromatic residues in the middle of the ACE2 binding surface (Fig. 2d)."

12. Page 7, lines 147-148. "At 2 dpi, the infectious virus titre (TCID₅₀) in nasal wash was significantly decreased when compared to controls in time- and dose-dependent manners" It is not clear what is being compared here - all samples or only CPT-P59 at 30mg/kg. The first P value is barely if not even significant.

Response: We thank reviewer #1 for this insightful comment and have now revised the sentence to clarify the meaning.

"At 2 dpi, the infectious virus titre (TCID₅₀) in nasal wash was significantly decreased when compared to controls in time- and dose-dependent manners, and the infectious virus was not detected at 6 dpi in animals treated with 30 mg/kg of CT-P59."

→

"At 2 dpi, the infectious virus titre (TCID₅₀) in nasal wash was significantly decreased in animals treated with 30 mg/kg of CT-P59 when compared to control group, and the infectious virus was not detected at 6 dpi"

13. Page 8, line 159. Please delete "the".

Response: We have deleted "the".

14. Page 8, line 174. "45 and 90 mg/kg" It would have been helpful to also have done lower amounts.

Response: We agree with reviewer's comment. The efficacy testing with lower amount of CT-P59 in the monkey study would be helpful to determine the optimal dose for the clinical trial, considering that no replicating virus was detected in upper airway in animals given 45 mg/kg of CT-P59 at 2 dpi. However, although the neutralizing effect at lower dose was not determined in monkey experiment, we expect that it is likely to be effective, based on the results from hamster and ferret experiments. Furthermore, clinically efficacious doses were estimated to be 45 mg/kg and 90 mg/kg based on the monkey study, a dose including extra precaution to minimize the treatment failure in high risk patients. It's because the data on the pathogenesis of COVID-19 and therapeutic potential of neutralizing antibody is not sufficiently accumulated.

15. Page 9, lines 190-193. "Unlike RBD-targeting antibodies with similar sensitivities to the viral neutralization on a D614G variant²¹, CT-P59 can effectively neutralize D614G variant; the underlying mechanism remains to be elucidated in terms of S protein stability and viral kinetics"

I don't really know what the sentence means. The D614G mutation is not in the RBD or RBM. It is good to know that there is no real difference in neutralization, but I am not sure what underlying mechanism is being referred to here.

Response: We appreciate the reviewer's comment. We re-wrote the sentence as below.

"Importantly, CT-P59 significantly inhibited the viral replication of clinical isolates by *in vitro* PRNT. Unlike RBD-targeting antibodies with similar sensitivities to the viral neutralization on a D614G variant²¹, CT-P59 can effectively neutralize D614G variant; the underlying mechanism remains to be elucidated in terms of S protein stability and viral kinetics."

→

"Importantly, CT-P59 significantly inhibited the viral replication of clinical isolates, wild type and D614G variant by *in vitro* PRNT."

16. Page 11, lines 227-229. "We therefore propose that CT-P59 may have more chance to block ACE2-RBD interaction in pre-fusion state than IGHV3 group antibodies if there is slight hinge-like movement in RBD region". It is not known what is meant by pre-fusion state here- it cannot be just all RBD down as cryo-electron tomography has shown up RBD up states also on viruses. All of the states alluded to here seem to be pre-fusion.

Response: We agree that using a term of "pre-fusion state" is confusing. What we were trying to propose was that CT-P59 may more readily bind to RBD trimer than IGHV3 group antibodies when RBD undergoes conformational change from "down" state to "up" state since CT-P59 has much fewer steric clash than IGHV3 group antibodies. However, while we were comparing with other neutralizing antibodies that the reviewer#1 suggested, we have found that COVA2-39 adopts similar RBD binding angle with CT-P59 in Supplementary Fig. 3a although its orientation sitting on RBD is different from CT-P59 as shown in Supplementary Fig. 3b. Because COVA2-39 is belonged to IGHV3 germ line, we cannot argue all IGHV3 antibodies have heavy clashes with adjacent RBD protomer on "down" conformation. Thus, we have removed the structural comparison with IGHV3 group antibodies as well as P2B-2Fb as below.

"Structural alignment of each group of neutralizing antibodies with the "down" form of SARS-CoV-2 S protein trimers showed that the IGHV3 antibody group heavily clashes with the adjacent RBD protomer, whereas P2B-2F6 can bind to the trimer without any collision with the adjacent molecules (Extended Data Fig. 4b). Although CT-P59 collides with the Asn343 glycosylated site on adjacent protomer, it has much fewer clashes compared with IGHV3 group antibodies. All the evaluated antibodies can interact with RBD on the "up" conformation without any steric hindrance. We therefore propose that CT-P59 may have more chance to block ACE2-RBD interaction in pre-fusion state than IGHV3 group antibodies if there is slight hinge-like movement in RBD region."

→

"Structural alignment of CT-P59 with SARS-CoV-2 S protein trimers showed that CT-P59 binds to RBD on the "up" conformation without any steric hindrance whereas it collides with the Asn343 glycosylated site on adjacent protomer in the "down" form (Supplementary Fig. 3c)."

17. Page 11, line 239-240. "The early clearance of infectious virus suggests that CT-P59 might be an option for COVID-19 patients as combination therapy". How does CT-P59 compare with other antibodies that have reported animal studies?

Response: Therapeutic option of CT-P59 as a combination therapy was considered to be delivered with the Standard of Care, not other developing neutralizing antibody drugs, because there have not been an authorized neutralizing antibody for treatment of SARS-CoV-2 yet. For the comparison to other developing antibodies on in vivo efficacy through the assessment from literature publicly available, it was demonstrated that CT-P59 shows similar neutralizing effect in hamster and/or NHP models, although dose was not the same.

Regeneron –REGN-COV2 antibody

-The antibodies, known as REGN-COV2, are a cocktail of two potent neutralizing antibodies (REGN10987+REGN10933) targeting non-overlapping epitopes on the SARS-CoV-2 spike protein. Six monkeys were given REGN-COV2 (mAb) and six were given placebo, through intravenous administration. The monkeys were then challenged with virus through intranasal and intratracheal routes three days after being dosed with antibodies. For animals receiving REGN-COV2 prophylaxis was observed accelerated clearance of genomic RNA (gRNA) with almost ablation of subgenomic RNA (sgRNA) in the majority of the animals. In macaques treated with the drug one day after infection, the authors reported faster viral clearance than in controls who'd not been treated. Next, hamsters treated with the drug two days before infection exhibited "dramatic protection from weight loss" and decreased viral load in the lungs. They also reported benefits for hamsters treated one day after infection, as compared to controls. (Baum et al., 2020; REGN-COV2 antibodies prevent and treat SARS-CoV-2 infection in rhesus macaques and hamsters)

Shanghai Junshi Biosciences – CB6 antibody

The antibody, known as CB6 mAb, a neutralizing mAb isolated from a convalescent COVID-19 patient, the interacting epitopes on SARS-CoV-2-RBD for CB6 are highly overlapped with the binding epitopes of hACE2. Nine male rhesus macaques were divided into pre-challenge (prophylactic), post-challenge (treatment) and control groups with 3 animals in each group. Before infection, the animals of pre-challenge group were infused with 50 mg/kg CB6-LALA intravenously. One day later, all macaques were inoculated with 1×10^5 TCID50 SARS-CoV-2 via intratracheal intubation. While the post-challenge group were also infused with 50 mg/kg antibody CB6-LALA on days 1 and 3 post challenge and three monkeys in the control group were given PBS as a control. Viral RNA loads in throat swabs determined by qRT-PCR were monitored for 7 days. The authors reported strong prophylactic and therapeutic effect of CB6-LALA antibody against SARS-CoV-2 infection in both viral load (throat swabs). (Liu et al., 2020; A human neutralizing antibody targets the receptor binding site of SARS-CoV-2)

18. Page 12, lines 245-247. "Therefore, these observations suggest that CT-P59 can remarkably neutralize SARS-CoV-2 via binding to RBD and ameliorate pathological symptoms without ADE during clinical trials". The lack of any ADE is important to show, but 'remarkably' is far too strong and should be removed.

Response: We have deleted “remarkably”.

19. Page 29, lines 584-585, Reference 43. Different font from the rest of the text.

Response: We have made same font for all references.

20. Extended Data Table 2 | Data collection and refinement statistics

1) Please add number of unique reflections measured, Rpim, CC1/2, Wilson B.

2) α , β , γ ($^\circ$) Please replace “90.00, 90.00, 90.00” with “90 90 90”.

3) $I / \sigma I$ “12.44 (1.63)” 1 decimal point is sufficient.

4) Completeness (%) “99.78 (99.88)” 1 decimal point is sufficient.

5) Rwork / Rfree “0.2167 / 0.2420” 3 decimal points are sufficient. B-values, please truncate to integers, decimal points are not meaningful.

Response: We have included unique reflections measured, Rpim, CC1/2, Wilson B values in Supplementary Table 2. In addition, the decimal points of some values were changed according to reviewer’s suggestions.

21. Page 4, Extended Data Fig. 1, line 5. Please replace “Data is” with “Data are”.

Response: We have replaced “Data is” with “Data are”.

Reviewer #2 (Viral immunity) (Remarks to the Author)

Kim et al. report a therapeutic mAb against SARS-CoV-2 spike RBD. The study is comprehensive with results from virology, biophysics, structure, and animal efficacy. The manuscript is well written. This reviewer supports the publication in Nature Communications with some minor modifications.

1. For the in vivo efficacy results on respiratory samples, the authors used infections virus amounts as an efficacy readout. This could be complicated with administered mAb that may inhibit the infectious virus during neutralization assay. So, it would be important to test one set of these samples using RT-PCR method.

Response: We thank reviewer #2 for this insightful comment and, as suggested by this reviewer comment, we have added data of viral RNA level for respiratory tract by using PCR method. Viral RNA copy numbers in upper respiratory tract were measured from nasal wash samples for ferrets, and viral RNA in lower respiratory tract were measured from lung tissues for ferrets,

hamsters, and NHP. In addition, infectious viruses were measured by TCID assay from throat swab for NHP.

We have re-organized the figures for viral loads throughout Fig. 3-4 and Supplementary Fig. 5, and now there are several changes in the number of figures as below;

1. Figure 3 has changed for data on viral load from upper respiratory tract of ferrets and NHP.
 - A. Fig 3b has newly added for the level of viral RNA in nasal wash of ferret; former Fig. 3b for the level of viral RNA in rectal swab of ferret has moved to Supplementary Fig. 5a.
 - B. Fig 3d has newly added for the level of infectious virus in throat of NHP; former Fig. 3d for the level of viral RNA in rectal swab of NHP has moved to Supplementary Fig. 5b.
2. Figure 4 has changed for data on viral load from lower respiratory tract of ferrets, hamsters, and NHP.
 - A. Fig 4a, b, and c for the level of replicating virus in lungs, have moved from former Extended data Fig 6a, b, and c.
 - B. Fig 4d, e, and f have newly added for the level of viral RNA in lungs of ferrets, hamsters, and NHP, respectively
3. Supplementary Fig. 5 (former Fig. 3) contains the data on the level of viral RNA in GI tract.
 - A. Supplementary Fig 5a, b have moved from former Fig 3b, d, respectively.

2. The order of Extended Data figures should be reorganized in the order of their appearance in text.

Response: We have reorganized the order of Extended Data figures in the order of their appearance in text. “Extended Data figures” were re-named as “Supplementary figures” as per formatting instruction of *Nature Communications*.

3. Lines 104-105: the Kd value of 27 pm does not match “ $2.51 \times 10^{-10} \text{ M}$ ” in Extended Data Table 1.

Response: Please see our response for Reviewer1 #4

4. Remove “novel” in the manuscript title.

Response: We have replaced “novel” with “therapeutic” in the title.

REVIEWERS' COMMENTS

Reviewer #1 (Remarks to the Author):

All the previous comments were addressed – thanks. Some other minor suggestions include:

1. Page 10, lines 214-215: “The neutralizing antibody P2B-2F6 which is based on the IGHV4-38 gene”. However, this antibody is likely to be more precisely encoded by IGHV4-38-2 instead of IGHV4-38 based on IgBLAST analysis.
2. Likewise, in Supplementary Fig. 3a, the genes encoding EY6A, H014, P2B-2F6, and BD23 are more likely to be IGHV3-30-3, IGHV1-69-2, IGHV4-38-2, and IGHV7-4-1 instead of IGHV3-30, IGHV1-69, IGHV4-38, and IGHV7-4 that are labeled in the figure.
3. In the figure legend for Supplementary Fig. 3a, it was shown that “gray surface model highlighted with ACE2 interaction region in red”. It looks like the red surface represents the epitope instead of the ACE2 interaction region.
4. In the figure legend of Fig. 2c, please specify the structure used for the calculation of ACE2-binding residues.

Reviewer #2 (Remarks to the Author):

The authors have addressed this reviewer's comments.

<Point-by-point responses to reviewers' comments>

We gratefully appreciate the time and effort on the part of the reviewers in providing critical comments to improve the manuscript. In our revised version we have endeavored to address each of the concerns raised in the review as listed below.

Reviewer #1 (Remarks to the Author)

All the previous comments were addressed – thanks. Some other minor suggestions include:

1. Page 10, lines 214-215: “The neutralizing antibody P2B-2F6 which is based on the IGHV4-38 gene”. However, this antibody is likely to be more precisely encoded by IGHV4-38-2 instead of IGHV4-38 based on IgBLAST analysis.

Response: We have revised “IGHV4-38” to “IGHV4-38-2”.

2. Likewise, in Supplementary Fig. 3a, the genes encoding EY6A, H014, P2B-2F6, and BD23 are more likely to be IGHV3-30-3, IGHV1-69-2, IGHV4-38-2, and IGHV7-4-1 instead of IGHV3-30, IGHV1-69, IGHV4-38, and IGHV7-4 that are labeled in the figure.

Response: We have revised the germline in the Supplementary Fig. 3a as reviewer#1 suggested.

3. In the figure legend for Supplementary Fig. 3a, it was shown that “gray surface model highlighted with ACE2 interaction region in red”. It looks like the red surface represents the epitope instead of the ACE2 interaction region.

Response: We appreciate reviewer#1’s thorough comment. We have corrected “ACE2 interaction region” to “epitope region” in the figure legend for Supplementary Fig. 3a.

4. In the figure legend of Fig. 2c, please specify the structure used for the calculation of ACE2-binding residues.

Response: We have specified the PDB code (6LZG) used for the calculation of ACE2-binding residues in the figure legend of Fig. 2c.

Reviewer #2 (Remarks to the Author)

The authors have addressed this reviewer's comments.

Response: None.